Manuscript prepared for Nat. Hazards Earth Syst. Sci.
with version 2014/09/16 7.15 Copernicus papers of the LaTeX class copernicus.cls.
Date: 29 May 2017

# Simple and approximate estimation of future precipitation return-values

Rasmus E. Benestad[1], Kajsa M. Parding[1], Abdelkader Mezghani[1], and Anita V. Dyrrdal[1]

[1]The Norwegian Meteorological Institute, Henrik Mohns Plass 1, Oslo, 0313, Norway

*Correspondence to:* Rasmus E. Benestad (rasmus.benestad@met.no)

**Abstract.** We present estimates of future twenty-year return-values for 24-hr precipitation based on multi-model ensembles of temperature projections and a crude method to quantify how warmer conditions may influence the precipitation intensity. Our results suggest an increase by as much as 40–50% projected for 2100 for a number of locations in Europe, assuming the high RCP8.5 emis-

sion scenario. The new strategy was based on combining physical understanding with the limited available information, and utilised the covariance between the mean seasonal variations in precipitation intensity and the North Atlantic saturation vapour pressure. Rather than estimating the expected values and interannual variability, we tried to estimate an "upper bound" for the response in the precipitation intensity based on the assumption that the seasonal variations in the precipitation intensity

are caused by the seasonal variations in temperature. Return-values were subsequently derived based on the estimated precipitation intensity through a simple and approximate scheme that combined the one-year 24-hr precipitation return-value and downscaled annual wet-day mean precipitation for a 1-in-20 year event. The latter was based on the $95^{th}$ percentile of multi-model ensemble spread of downscaled climate model results. We found geographical variations in the shape of the seasonal

cycle of the wet-day mean precipitation which suggest that different rain-producing mechanisms dominate in different regions. These differences indicate that the simple method used here to estimate the response of precipitation intensity to temperature was more appropriate for convective precipitation than for orographic rainfall.

## 1 Introduction

Extreme precipitation is associated with flooding and landslides and can have detrimental effects on infrastructure and society (Trenberth et al., 2003), as for example during the unusually intense cloud-

burst in central Copenhagen on July 2, 2011 which caused massive flooding, and the 2002 floods in central and eastern Europe (Hov et al., 2013). Return-values are commonly used in planning and design of weather-resilient infrastructure by quantifying the magnitude of a typical extreme event.

However, the return-values are not stationary, and according to the reinsurance company Munich Re (Hov et al., 2013), there has been an increase in the annual number of loss events related to weather. Assessments carried out by the Intergovernmental Panel on Climate Change (IPCC) indicate that heavy precipitation will become more severe in already wet areas in the future (Stocker, T.F. et al., 2013; Field et al., 2012). These assessments have largely been based on global climate model (GCM)

output and have not made use of additional local information such as meteorological observations. One of the difficulties of using observational data is the patchy character of the information because of missing data and short records. GCMs are not designed to represent local precipitation statistics corresponding to rain gauge data, but are expected to reproduce the nature of large-scale (regional and global) phenomena and processes seen in the atmosphere and oceans. Also, some elements are

reproduced with higher skill than others. In other words, GCMs provide a more reliable picture of the temperature aggregated over larger spatial scales than of grid-box precipitation estimates (Takayabu et al., 2015), and their ability to simulate large-scale features can be utilised for inferring changes to local precipitation through downscaling (Benestad, 2008). This caveat also applies to regional climate models (RCMs), which too have a minimum skillful scale (Takayabu et al., 2015), and have

a limited ability to reproduce the observed precipitation statistics (Orskaug et al., 2011; Benestad and Haugen, 2007). Nevertheless, RCMs have been used to study precipitation extremes (Frei et al., 2006, e.g.), although the heavy computational demands have limited the analysis to a small number of GCMs which means that the ensembles do not provide a realistic range of possible outcomes associated with natural variability and model uncertainty (Deser et al., 2012).

Traditional methods of estimating return-values that make use of the extreme value theory (EVT) are sensitive to sampling fluctuations and require long data records to avoid extrapolation of the extreme characteristics (Coles, 2001; Papalexiou and Koutsoyiannis, 2013). Extreme precipitation modeled through EVT usually describes amounts that are far out in the tail of the distribution and associated with low probability, and the estimates may change when new extremes are sampled.

Most uses of EVT also assume stationarity, although there are ways to account for trends (Cheng et al., 2014).

Local precipitation has been notoriously difficult to predict (Stocker, T.F. et al., 2013; Field et al., 2012; Arkin et al., 1994), and one reason may be that it has involved quantities such as the monthly mean precipitation that are calculated from a blend of different (both dry and wet days) conditions

and phenomena without accounting for these differences. There are many different types of phenomena that generate precipitation, e.g., the formation of stratonimbus, mid-latitude cyclones, fronts, atmospheric rivers, convection, as well as warm and cold initiation of rain (Fleagle and Businger, 1980; Berg et al., 2013; Trenberth et al., 2003). Some of these are more strongly present in certain

regions and seasons. For instance, convective precipitation is typically a summer phenomenon at mid-to-high latitudes, whereas mid-latitude cyclones are more pronounced in autumn, winter, and spring. Another reason for the limited success may be the small sample size in calculations of the mean precipitation for locations and seasons where it rains rarely. For example, if it rains less than 30% of the total number of days in a month, the monthly average precipitation is based on less than 10 values. The quantification of future extreme precipitation is associated with uncertainties from a number of sources, e.g., model imperfections, sparsity of data, sensitivity to random variations in small samples constituting the tail of the distribution, and non-stationarity, as well as the representation of natural variability. Large multi-model ensembles can be used to explore the natural variability of the climate system, although the range of the ensembles also includes other sources of uncertainty and variability, and some ensemble members may be inter-dependent (Sanderson et al., 2015).

Moderate extremes in 24-hr precipitation amounts ($X$) can be approximated with an exponential distribution (Benestad et al., 2012a, b; Benestad, 2013), which is described with one parameter - the wet-day mean $\mu$ - and its percentile ($q_p$) can be estimated as $q_p = -\ln(1-p)\mu$. The exponential distribution can be used to estimate changes in the moderate upper tail of the statistical distribution, assuming that these follow changes in the bulk characteristics where the probability adds up to unity (Benestad and Mezghani, 2015). This approximation has been tested against daily rain-gauge records from around the world, confirming that the exponential distribution ($q_p = -\ln(1-p)\mu$) predicts the observed precipitation percentiles with high accuracy for low to moderately heavy precipitation amounts (Figure SM1). This means that $\mu$ is useful for risk analysis, to estimate upper percentiles of 24-hr precipitation amounts, because the $95^{t}h$ percentile $q_{95}$ is expected to change proportionally with $\mu$ (Benestad, 2013; Benestad and Mezghani, 2015).

## 2  Data and Method

Our objective was to get estimates of future extreme precipitation that were robust to outliers in situations when local observations are limited and to avoid some of the caveats described above. We therefore explored a method of extracting information about extreme precipitation from the multitude of data sources available while reducing the uncertainty associated with small sample sizes and blended conditions. Our analysis drew on available and relevant information concerning precipitation, for instance geographical variations, seasonal variations, ensemble spread, and different physical processes present during wet and dry days, respectively.

The estimated precipitation change was based on the change in temperature and did not explicitly take atmospheric circulation changes or feedback processes into consideration. This change can for all intents and purposes be interpreted as a zeroth-order measure of an"upper bound" of change in precipitation intensity associated with increased temperature, rather than the most likely value. Attributing all of the seasonal variations in the precipitation intensity to its covariance with temper-

ature may inflate the role of the temperature, as other factors exhibit a similar mean seasonal cycle
and may have an influence on the precipitation intensity. For this reason, we use the terms "upper
bound" and "potential sensitivity". It is also true that other unaccounted-for processes possibly may
influence the precipitation intensity in a nonlinear fashion and possibly result in even higher inten-
sities if they also change in the future. However, as long as (a) such factors have an approximately
linear dependency on the temperature and (b) the temperature may be taken as a proxy for climate
change, then this simple assumption may provide a reasonable figure. This simple method differs
from traditional methods in that rather than attempting to specify the *most likely* value, it estimates a
kind of *upper bound* of the systematic response of extreme precipitation to changes in temperature.
We henceforth describe this relation as the *potential sensitivity* (PS) since the calibration used the
covariance of the mean annual variation that may exaggerate the effect of the temperature. This is
described in more details below.

Our approach was based on empirical-statistical downscaling (ESD) applied to a large multi-
model ensemble to provide estimates of return-values for heavy precipitation, and is an alternative
to EVT-based approaches. It provided an estimate that was more approximate and crude, but less
sensitive to outliers because a larger portion of the data sample is used.

The supporting material (SM) provides more details and explanations of the strategy, as well as
the R-scripts used to perform the analysis. The calculations and graphics were produced with the
open-source R-package 'esd' (Benestad et al., 2015).

## 2.1 Data

Precipitation observations were obtained from the daily ECA&D dataset (Klein Tank et al., 2002)
for 1032 stations in northern Europe with data available for the time period 1961–2014 (Figure 2).
Surface temperature data from the NCEP/NCAR reanalysis 1 (Kalnay et al., 1996) over a selected
North Atlantic domain (100°W-30°E/0°N-40°N; see Figure SM2) were used to calculate the pre-
dictors for the downscaling, and corresponding projections from the CMIP5 ensembles of GCMs
assuming the RCP 2.6, 4.5, and 8.5 scenarios (**?**) were used for the projections of future change
(Table 1). We used the NCEP/NCAR reanalysis 1 because the data covered the 1961–2014 period
and because it provided a representation for the surface temperature that was comparable to that of
the CMIP5 GCMs.

## 2.2 Downscaling method

### 2.2.1 Predictand: the annual wet-day mean precipitation

A traditional approach for modeling and analysing precipitation typically involves the monthly mean
precipitation ($\bar{X}$), but in this study, we instead downscaled the wet-day mean, $\mu$. In this analysis we
used $\mu$ to represent the wet-day mean precipitation in general, reflecting both the annual wet-day

mean precipitation and the mean seasonal variations in the wet-day mean precipitation estimated for the 12 calendar months. The mean precipitation was not the optimal quantity for describing

precipitation statistics because in most places, it doesn't rain every day, and the proportion of wet days to total number of days in a monthly sample can have implications for the estimation of the statistical parameters describing the distribution. The mean precipitation can be expressed as the product of the wet-day frequency ($f_w$) and $\mu$ according to $\bar{X} = f_w\mu$. A comparison between the seasonal dependence of $\bar{X}$, $\mu$, and wet-day frequency $f_w$ indicated a stronger seasonal cycle in $\mu$

than in $f_w$ and $\bar{X}$ (see Figure SM3). The weaker seasonal cycle in $\bar{X}$ was due to the blending of different types of weather conditions in the mean precipitation. The strong seasonal cycle of $\mu$ indicated a sensitivity to climatological variations, which is an important requirement for the statistical downscaling strategy proposed here.

### 2.2.2 Predictor: the saturation vapour pressure

We assumed that the vapour saturation pressure, $e_s$, is more linearly related to the atmospheric water content and precipitation than the temperature, and hence used $e_s$ as a predictor in the downscaling of the annual wet-day mean precipitation $\mu$ (Fujibe, 2013; Pall et al., 2007; Benestad and Mezghani, 2015). The saturation vapour pressure was estimated from the surface temperature (0.995 sigma level), $T$.

$$e_s = 10^{(11.40-2353/T)} \tag{1}$$

This approximation was based on integration of the Clausius-Clapeyron equation, assuming a constant latent heat of vaporisation (see Equation 2.89 in Fleagle and Businger (1980)). The mean seasonal variations in the regional average $e_s$ over the North Atlantic domain was used as predictor for $\mu$ , based on its mean seasonal variation (Figure 1) and the motivation was that it can be con-

sidered as the source region for humidity in Europe. The domain was set after some trials for a few test stations, but no systematic study or tuning of the predictor domain was conducted. The predictor index was calculated from gridded temperature data from reanalyses and global climate models (GCMs) and then spatially and temporally aggregated, where monthly gridded $e_s$ values were estimated according to equation 1 and surface temperatures from the multi-model ensemble and used to

downscale an ensemble of local results of annual wet-day mean precipitation $\hat{\mu}$ (here $\hat{\mu}$ is used for predicted annual mean).

### 2.2.3 The empirical-statistical model

A model for predicting the annual wet-day mean precipitation $\hat{\mu}$ can be constructed as a sum of a constant, $\beta_0$, a term depending on the saturation vapour pressure, $\beta_T e_s$, and a Gaussian noise term,

$N(0,\sigma)$, assuming that factors other than temperature that are affecting wet-day precipitation are stochastic and stationary:

$$\hat{\mu} = \beta_0 + \beta_T e_s + N(0,\sigma). \tag{2}$$

The assumptions about other factors being stationary and stochastic is partly based on the heuristic notion of physical interdependencies between various aspects of the planetary atmosphere in general
and that the temperature is a proxy for such influences. One example may be the cloud top height which is expected to be influenced by the convective available potential energy (CAPE) that is sensitive to temperatures. We used the observed standard deviation of $\mu$ in the month with the highest inter-annual variability as an estimate of the standard deviation $\sigma$ of the noise term $N$, which in this case was August. We calculated the coefficients $\beta_0$ and $\beta_T$ by linear regression between the mean
seasonal cycle of the observed monthly mean $\mu$ and the corresponding seasonal cycle of the regionally averaged $e_s$ calculated from reanalysis temperature data from the Atlantic domain, as described in Section 2.2.2. The coefficient $\beta_T$ is the scaling ratio which we refer to as the potential sensitivity.

Annual mean time series $\hat{\mu}$ were then derived by applying the downscaling models to annual mean $e_s$ time series obtained from reanalysis or GCM temperature data from the same domain.
The GCM results were not bias-adjusted, however, the use of large-scale ($100°$W-$30°$E/$0°$N-$40°$N) spatially and annually aggregated mean helped mitigating the effects from systematic model biases. The model represented an approximation of the systematic effect that temperature changes can have on $\mu$, rather than a most likely value. It is possible that other factors that play a role in precipitation also exhibit a seasonal cycle and interfere with the regression analysis so that the coefficient is
weaker or stronger than the true influence of temperature on precipitation.

A 90% uncertainty range for $\hat{\mu}$ was estimated for the projections based on the ensembles of downscaled results, taken as the limit between the $5^{th}$ and $95^{th}$ percentiles (see, e.g, Figure SM4). This interval included the noise term $N(0,\sigma)$, and captured the observed year-to-year variations as well as model differences (Deser et al., 2012). We assumed that the multi-model ensemble spread for any
given year could approximately represent the typical year-to-year variance, which meant that the $95^{th}$ percentile for $\hat{\mu}$, which we henceforth refer to as $\hat{\mu}_{95}$, could be used as a proxy for the value to be exceeded once in 20 years (Benestad, 2011). (The 1-in-20 year event has a probability of 0.05 (1/20) of occurring in a given year, and the $95^{th}$ percentile represents a limit that only 5% (1 in 20) of the distribution exceeds.)

## 2.3 Return-value probabilities

To estimate future return-values based on the downscaled $\hat{\mu}$, we again assumed that the wet-day precipitation-amount was exponentially distributed and that the probability for 24-hr precipitation exceeding a critical threshold $x$ could be calculated as follows:

$$Pr(X > x) \approx f_w e^{-x/\mu}, \tag{3}$$

where $f_w$ was the wet-day frequency (Benestad and Mezghani, 2015). Previous analysis suggest that the exponential distribution gives a reasonable description of the probabilities for moderate precipitation events such as the 95-percentile, but is not expected to be suitable for rare extremes much beyond the 20-year return level (Benestad, 2013).

The probability associated with the one-year return-value of 24-hr precipitation is approximately $Pr(X > x) = 1/365.25$, and the corresponding threshold value was approximated according to

$$x_{1yr} \approx \mu \ln(365.25 \ f_w). \tag{4}$$

Previous comparison between the return-values based on Equation 4 and general extreme value theory, has suggested that they give roughly similar results (Benestad and Mezghani, 2015). A test of Equation 4 indicated that the return-values scale with $\mu$: values of $x_{1yr}$ that were associated with high percentiles and low values of $\hat{\mu}$ approximately corresponded to $x_{1yr}$ with low percentiles and high values of $\hat{\mu}$ (Figure **??**). Based on Equation 4, we made a rough estimate of the 20-year return-value for the 24-hr precipitation amount ($x_{20yr}$) by replacing $\mu$ with the 20-year return-value of the annual wet-day mean. The estimate for $\hat{x}_{20yr}$ was calculated based on the downscaled annual wet-day mean precipitation, using the $95^{th}$ percentile $\hat{\mu}_{95}$ as a proxy for the 20 year return values:

$$\hat{x}_{20yr} = \hat{\mu}_{95} \ln(365.25 \ f_w). \tag{5}$$

In calculating future return-values, we neglected changes in $f_w$ and simply assumed that it will remain constant. Previous analysis has indicated that the wet-day frequency is strongly influenced by circulation patterns (Benestad and Mezghani, 2015), and that it is closely connected to slow natural variations such as the North Atlantic Oscillation (NAO) (Hurrell, 1995). Such natural variations are difficult to predict and there is little evidence of a systematic shift in the frequency of different circulation patterns.

## 2.4 Principle component analysis of the seasonal cycle

Principal component analysis (PCA) was used to extract the most dominant shapes of the seasonal cycle in $\mu$ amongst the observation sites (2). The mean seasonal cycle was estimated for each site

and used to construct a data matrix $X$ with 12 columns (one for each month) and $n$ rows (one for each site). Singular value decomposition (SVD) was then used to compute the principal components: $U\Sigma V^T = X$, where $U$ is the left inverse, $V$ the right inverse, and $\Sigma$ is a diagonal matrix holding the eigenvalues (Press et al., 1989; Strang, 1988). The procedure deconstructed the data into a set of shapes of the seasonal cycle, corresponding eigenvalues that described the explained variance, and a

spatial matrix that described the relative strength of each shape at the different locations.

## 3   Results and discussion

### 3.1   Potential sensitivity and the seasonal cycles in $\mu$ and $e_s$

The mean seasonal cycles of $\mu$ at many European locations co-varied with the mean seasonal cycle of $e_s$ in the North Atlantic domain. This can be seen as a validation of the assumptions underlying

the empirical model, because the downscaling models were based on the regression between the seasonal cycles of $e_s$ and $\mu$ (Equation 2). Figure 1 provides an example of a scatter plot between the mean seasonal variations in $e_s$ (x-axis) and the corresponding cycle in $\mu$ (y-axis) for one location (Velikie Luki, Russia). The example in Figure 1 was not unique: there was a high and statistically significant correlation ($R^2 > 0.6$; Figure SM5) between the seasonal cycle of these two quantities

for many of the rain gauge records (612 of the 1032 stations). The majority of the locations with a poor fit ($R^2 < 0.6$) were found along the Norwegian west coast and southeast of the Alps, while inland sites and locations in central Europe had higher $R^2$ values (see Figure 2 where the size of the markers is proportional to $R^2$). This indicated that a linear relationship between $\mu$ and $e_s$ could not be expected in regions where orographic precipitation was dominant. Downscaled projections were

carried out only for the locations with a good fit ($R^2 > 0.6$).

It was also evident that there were pronounced year-to-year variations in the wet-day mean (vertical error bars in Figure 1) which were not related to the temperature, suggesting that factors other than temperature also played a role in precipitation variations. The downscaling strategy adopted here was designed to evaluate the maximum potential effect of temperature changes on the wet-day

mean precipitation, and the scaling factor between the two is described as the potential sensitivity. Since other processes also influenced precipitation, the method could not be expected to reproduce past interannual variability, but it could be used to obtain a rough estimate of the effect of temperature changes on precipitation.

Figure 2 presents maps showing the two major components of the mean seasonal cycle in $\mu$,

which together accounted for 94% of the variability for the 1032 locations examined. The spatial patterns in the principle components (PC) revealed different seasonal cycles of precipitation along the mountainous western coast of Norway and close to the Alps compared to the rest of Europe, probably related to orographic effects. There was a gradient in the shape of the mean seasonal cycle in $\mu$ with the distance from the coast that was particularly visible over the Netherlands. Inland sites

indicated higher precipitation intensities during July and August, which could be associated with convective rainfall. We found a positive correlation between the spatial vector of the leading PCs and $R^2$ of the seasonal cycles of $e_s$ and $\mu$: 0.82 (with a 90% uncertainty range of 0.80, 0.84), but negative correlation for mode 2 (-0.84; -0.86,-0.82) and no significant correlation for mode 3 (0.00; -0.06, 0.06). This indicated that the dominant shapes of the seasonal cycle of $\mu$ in Europe were associated with a strong connection to the North Atlantic temperature.

## 3.2 Projections of future precipitation

Projected values of the annual mean wet-day mean, $\hat{\mu}$, based on the downscaling model (Equation 2) applied to the CMIP5 ensemble, are shown in Figure 3. The downscaled results suggested an increase of up to 13% in the wet-day mean from 2010 to 2100, assuming the RCP 4.5 emission scenario (Stocker, T.F. et al., 2013), and as much as 38% at many of the locations given the high emission scenario RCP8.5. The most extreme estimate was an 85% increase at Sihccajavri (Norway). Since the wet-day precipitation amount approximately followed an exponential distribution, the proportional change in any percentile was the same as for $\mu$. The insert in Figure 3 shows estimated changes for the emission scenarios RCP4.5, RCP2.6 and RCP 8.5, respectively, for both the ensemble mean and $95^{th}$ percentile.

An analysis of historical observations provided some indication of skill of the downscaling models in terms of predicting trends of $\mu$ based on the North Atlantic temperature (Figure SM6). The historical trends exhibited a more pronounced scatter than the predicted trends, suggesting that factors other than the sea surface temperature also had influenced the long-term changes. For most locations, there has been an increase in $\mu$ between 1961 and 2014, typically 0.1 mm/day per decade (Figure SM6–SM7).

Estimates of future 20-year return-values (Equation 5) based on $\hat{\mu}_{95}$ and assuming a constant value of the wet-day frequency, $f_w$, are shown in Table 2. Based on downscaling of the RCP4.5 scenario, the 20-year return values may increase by between 7% and 28% by 2100 (ensemble median: 11%), or assuming the high emission scenario RCP8.5, between 22% to 85% (ensemble median: 33%). Nevertheless, changes in $f_w$ may also influence the return-values, and an increase in the number of rainy days would imply an even stronger change in return-values.

The historical $f_w$ trends at the stations tend to cluster roughly around zero (Figure SM8). However, studying the geographical pattern of trends, we saw a general increase in southern Scandinavia and the Netherlands for the period 1961–2014, but a less coherent pattern elsewhere (Figure SM9). This implied that factors other than the North Atlantic temperature may also have played a role for past trends and future precipitation changes. The wet-day frequency was strongly influenced by the circulation patterns (Benestad and Mezghani, 2015) and could potentially be predicted based on the mean sea-level pressure, but here we have focused on the influence of temperature changes on the precipitation.

### 3.3 Validation of results

In order to assess the veracity of our results, we performed an independent test to examine the dependency of $\mu$ to temperature, consisting of a regression analysis comparing the spatial variations of the mean of $\mu$ and $e_s$ calculated from local temperature measurements (Benestad, 2007) (see Figures SM10–SM11). The test was limited to locations where both temperature and precipitation observations were available and did not involve the regionally averaged temperature of the North Atlantic domain. The geographical variations in the relationship between $\mu$ and $e_s$ was consistent with the regression coefficients from the downscaling models (Equation 2, Figure 3) within the range of estimated error margins (Figure SM11). An exception was seen in stations located in western Norway and south of the Alps, where the seasonal cycle regression also showed a weak relationship between $\mu$ and $e_s$. The fact that the link between $\mu$ and $e_s$ was found in both time and space provided a stronger indicator of a physical link than if it were limited to only the time dimension.

### 4 Summary and conclusions

We changeproposehave proposed a novel and simple method for obtaining an approximate estimate of changes in the return-values for 24-hr precipitation caused by a temperature change, taking all precipitation relevant processes into account. This method made use of the information embedded in the seasonal cycle, physical conditions, and multi-model ensembles, to provide a rough estimate of the potential sensitivity of precipitation intensity to temperature. The results suggested that the zeroth-order estimate for an *upper bound* of the twenty-year return-value for many European locations increases by 40-50% by 2100 for the RCP8.5 scenario, rather than the exact or most likely value.

One of the benefits of the proposed strategy for downscaling $\mu$ is that the description of the seasonal cycle does not require long data records and hence may provide a means for estimating a zeroth-order value for the potential sensitivity and an "upper limit" to the change in rainfall statistics in regions with limited observations. This strategy can be used for other mid-latitude locations, but further analysis is needed to see if it is applicable to the monsoon regions where the temperature is at maximum before the rains start. An alternative approach could be to estimate future changes in $\mu$ based on downscaled local temperature from GCMs and a similar regression model as used in the test described above.

The approach was based on a set of assumptions: (a) the maximum seasonal mean response of the wet-day mean precipitation to the seasonal variations in temperature is represented by a proportional change, (b) the $95^{th}$ percentile of the annual wet-day mean precipitation from large multi-model ensembles (e.g., CMIP5) can be used to represent a 20-year event, and (c) the wet-day frequency is stationary. On the one hand, this new strategy is less rigorous than traditional extreme value statistics,

325 but on the other hand, it is more robust to outliers even in cases when the available information is limited.

Another potential weakness of the study is the use of the multi-model ensembles as a representation of natural climate variability. These "ensembles of opportunity" involve non-independent members and cannot really be considered as a random sample of data (Sanderson et al., 2015). How-
330 ever, internal variability dominates the variance on regional and local scales and gives a spread that is comparable to the observed variations even in single-model ensembles (Deser et al., 2012).

*Acknowledgements.* The methods and results produced for this paper were connected to research carried out for the H2020 EU-Circle (GA no653824), Nordforsk eSACP. The work was supported by the Norwegian Meteorological Institute. The data used are listed in the references, supporting material, and tables, and made available
335 at http://github.com/metno/esd_Rmarkdown/tree/master/paper58 with the R-scripts to reproduce the analysis.

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

**"Worst–case" fit based on seasonal variations**

Figure 1

**Figure 1.** A comparison between the mean seasonal cycle in the saturation vapour pressure (x-axis) and the wet-day mean (y-axis) for the site Velikie Luki, Russia. The error bars indicate two standard deviations of the year-to-year variations in the two variables. An insert show the standardised seasonal cycles, both variables peaking in July-August (red line = $e_s$, blue line = $\mu$).

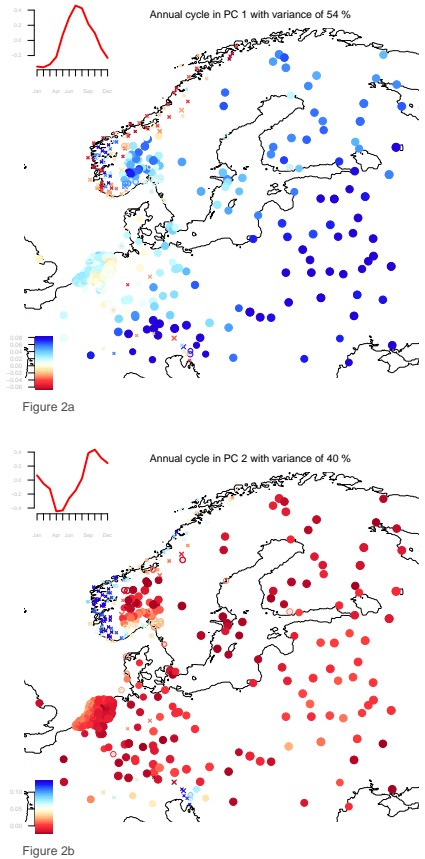

**Figure 2.** The weights for the two leading principal components (panels a and b) of the seasonal cycle of the wet-day mean precipitation $\mu$ in the 1032 rain gauge records. The colour of the symbols indicate how strongly the shape is present in the local seasonal cycle, and the size reflects $R^2$ from the regression analysis between $e_s$ and $\mu$ (see Figure SM5). Filled circle symbols were used for locations with $R^2 > 0.6$, empty rings $0.6 \geq R^2 > 0.4$, and crosses indicate locations with $R^2 < 0.4$. The shape of the seasonal cycle principal component for $\mu$ is shown in the insert (top right of each panel).

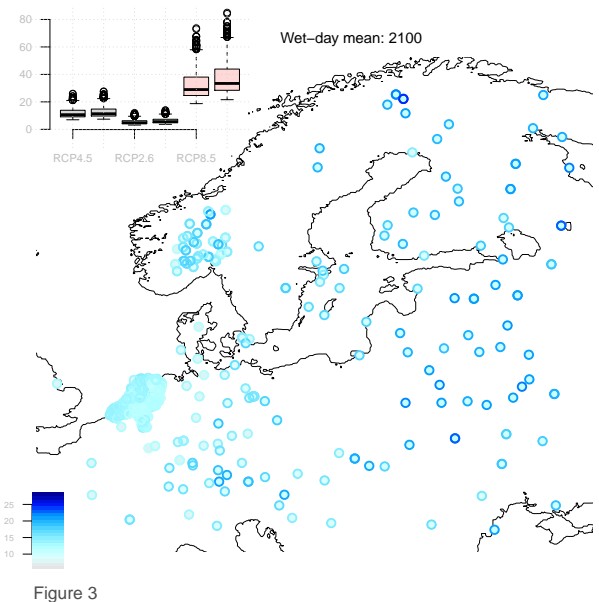

Figure 3

**Figure 3.** Projected local change from 2010 to 2100 in the ensemble mean and $95^{th}$ percentile annual mean $\mu$ for the RCP4.5 emission scenario. The colour of the inner part of the symbols indicate changes in the ensemble mean and the outer part the $95^{th}$ percentile in terms of percentages since 2010. The insert shows a boxplot of the projected change in $\mu$, both for the ensemble mean (left) and the $95^{th}$ percentile (right) of emission scenarios RCP4.5, RCP2.6, and RCP8.5, respectively.

**Table 1.** Summary of the CMIP5 experiments. The RCP4.5 was used as default here, whereas RCP2.6 and 8.5 were taken as lower and upper limits based on different emission scenarios.

| Ensemble | Total ensemble size (with duplicated models) |
|---|---|
| RCP4.5 | 108 runs |
| RCP2.6 | 81 runs |
| RCP8.5 | 65 runs |

**Table 2.** Summary of the projected change from 2010 to 2100 in the 20-year return-value for 24-hr precipitation under the assumption of stationary wet-day frequency. The sample comprises the 615 locations shown in Figure 3. The numbers represent the change in percentage with respect to year 2010.

| Ensemble | Min. | $q_{25}$ | Median | Mean | $q_{75}$ | Max. |
|---|---|---|---|---|---|---|
| RCP2.6 | 4% | 5% | 6% | 6% | 7% | 14% |
| RCP4.5 | 7% | 10% | 11% | 13% | 15% | 28% |
| RCP8.5 | 22% | 28% | 33% | 38% | 44% | 85% |

 **Supporting material - figure captions**

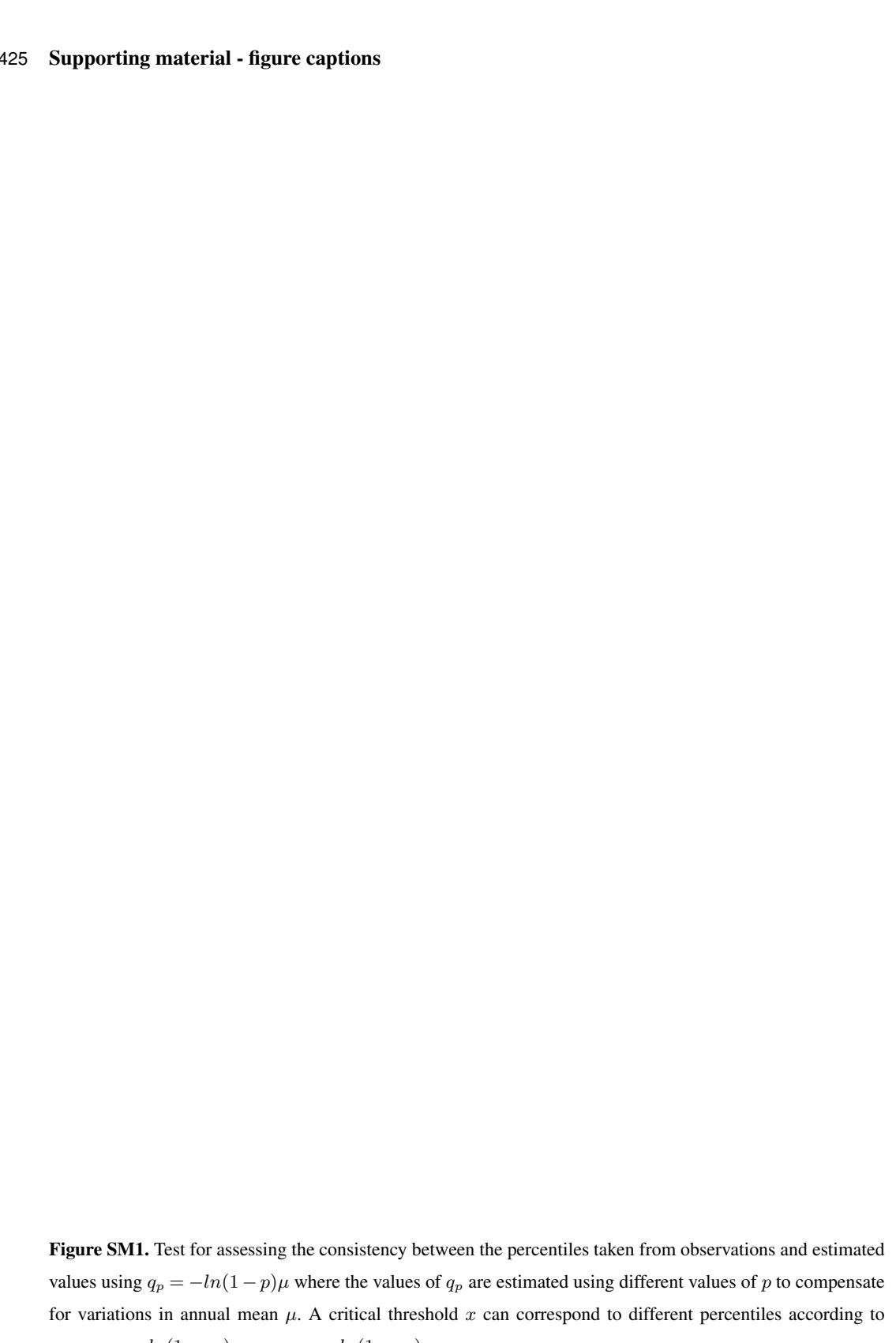

**Figure SM1.** Test for assessing the consistency between the percentiles taken from observations and estimated values using $q_p = -ln(1-p)\mu$ where the values of $q_p$ are estimated using different values of $p$ to compensate for variations in annual mean $\mu$. A critical threshold $x$ can correspond to different percentiles according to $x = q_{p1} = -ln(1-p_1)\mu_1 = q_{p2} = -ln(1-p_2)\mu_2$.

**Figure SM2.** The mean air temperature at 2m of the NCEP reanalysis data set over the chosen predictor domain 100W-30E/0N-40N.

**Figure SM3.** A comparison between the seasonal cycle in the mean precipitation, the wet-day mean precipitation, the wet-day frequency, as well as the wet and dry spell lengths. The most pronounced seasonal variations tends to be associated with the wet-day mean rather than the mean precipitation or the wet-day frequency.

**Figure SM4.** An example of projected annual wet-day mean precipitation $\mu$ for the three different emission scenarios RCP 4.5 (grey), RCP2.6 (green) and RCP8.5 (red), expressed as the relative change to the 2010 values (see Table 1).

**Figure SM5.** The statistics of the $R^2$ from the regression between the seasonal cycle in the the local wetday mean $\mu$ and the regionally averaged saturation vapour pressure es, estimated from the temperature over the seasonal cycles of the surface temperature over the North Atlantic domain (100W-30E/0N-40N; Figure SM3). There is a portion of stations with very low $R^2$ scores, but most stations suggest an explained variance exceeding 60%.

**Figure SM6.** A comparison between the longterm linear trends estimated from the observed annual mean $\mu$ and $\hat{\mu}$ values estimated with Equation 1 (see main manuscript) using the saturation water vapor es calculated from the NCEP temperature over the North Atlantic domain (100W-30E/0N-40N; Figure SM2). The scatter in the observed trends is greater than in the predicted ones, which is consistent with the wet-day mean also being affected by factors other than $e_s$.

**Figure SM7.** Map of the historical trends in the wet-day mean $\mu$ in the period 1961–2014. The trend is generally increasing, but there are a few stations showing a decrease. These outliers are probably spurious, as they do not match the bulk of the data.

**Figure SM8.** Trend estimates of the wet-day frequency $f_w$ for the 1032 locations for the period 1961–2014 suggests values scattered around zero. The cluster of trend values around zero is consistent with the annual wet-day frequency being stationary, but there are regions with significant trends (Figure SM9).

**Figure SM9.** Map of the historical trends in the wet-day frequency $f_w$ for the period 1961–2014. There has been a general increase in the number of wet-days in southern Scandinavia but otherwise no coherent pattern.

**Figure SM10.** Scatter plot showing the correlation between the climatological mean daily maximum temperature (converted to saturation vapour pressure) and the wet-day mean μ. The size of the symbols is proportional to the number of rainy days. Insert map shows locations of stations used to compare the climatological mean wet-day mean against the mean surface temperature. The colours of symbols in the scatter plot match those in the map. The data included CLARIS data set from South America, a subset of the ECA&D in Europe used in the COST-VALUE experiment 1, and a subset of station data from GDCN as in Smith et al. (2015) but selecting the stations with the longest records. The selection of location was also limited to sites where both temperature and precipitation had been recorded.

**Figure SM11.** Comparison between the regression coefficients estimated for each location based on the seasonal cycles in $\mu$ and $e_s$ (blue) and based on the regression analysis of the mean climatology of $\mu$ and $e_s$ at various stations in Europe, South America and North America as in Figure SM10 (grey). Error bars represent two standard errors. The size of the symbols is proportional to the $R^2$ statistics from the regression analysis between the two mean seasonal cycles. The comparison between the results from the two types of analyses suggests a consistency within the margin of error for the locations where the mean seasonal cycle in μ matched that of the regionally averaged es in the predictor domain (Figure SM2).