# Peer review of "Simple and approximate estimation of future precipitation return-values"

_Natural Hazards and Earth System Sciences, 2016_

## Referee Comment (RC1) · Anonymous Referee #1 · 13 Sep 2016

I think there is considerable substance in this paper, but confusing presentation and imprecise language gets in the way of making a convincing case for the approach that is proposed. This is apparent even from the title.  As soon as one combines the words "upper-limit" with "precipitation" in the same sentence, one evokes implicitly the idea of probable maximum precipitation – ie, an amount that is a physically plausible upper bound. The language in this paper is sufficiently imprecise that confusion about whether the aim is to estimate an upper bound for precipitation arises at several places. This is reinforced even within the abstract, where we read that the proposed method "utilises … to estimate the maximum effect that temperature change can have on precipitation …". Since the paper is about extreme precipitation, and the title refers to "upper-limit estimation", one could be excused for thinking that the interest is in estimating absolute upper bounds for 24-hour precipitation accumulations.

Awkward mathematical notation and poor conversion of the typescript to a pdf document also contribute confusion.  At various places the reader has to infill his/her own guess of what symbols are meant to be present in the text, apparently because they simply have disappeared, or have been misplaced, in the process that rendered the review document. More importantly, symbols such as the Greek letter $\mu$ seem to be used in multiple ways, with the reader needing to infer the correct interpretation from the context in which the symbol appears. For example, $\mu$ is used as the parameter of a probability distribution, as a quantity that varies from month to month, and as a quantile. It would be wonderful if standard statistical conventions could be used for notation. Generally, this means using Greek letters to represent unknown parameters that are to be estimated from observations, Greek letters with hats to denote parameter estimates, upper case Latin characters to represent random variables, and lower case Latin characters to represent realizations of those random variables (either observed or simulated), etc.).

The approach itself seems rather adhoc. The fundamental working assumptions are apparently that
   a) an exponential distribution fitted to all non-zero daily precipitation amounts will nevertheless represent most of the upper tail of the precipitation distribution reasonably well, including as far into the tail as 1-in-20-year 1-day events, and that
   b) precipitation frequency will not change.
The first, (a), is actually a very strong assumption given the existing body of observational evidence from multiple sources that suggests that extreme daily precipitation is heavy-tailed (that is, Frechet distributed rather than Gumbel distributed). A consequence of assuming the exponential distribution for daily precipitation is that block maxima (such as annual maxima) will converge to the Gumbel distribution (GEV distribution with shape parameter equal to zero) rather than the Frechet distribution.  The second assumption also seems a strong assumption given the sensitivity of the interpretation of changes in quantiles of nonzero precipitation amounts to changes in frequency (e.g., see Schar er al, 2016, doi: 10.1007/s10584-016-1669-2). To be fair, the authors defend (b) in this paper, and have discussed (a) in previous papers. Nevertheless, these are strong assumptions that work against the claim that the method provides robust estimates of an upper (uncertainty) bound for 20-year return values.

A third key assumption is that it is assumed that
   c) to the extent that 20-year return values are sensitive to temperature, uncertainty in projected changes in 20-year return values can be bounded by using the 95th percentile of changes in the parameter of the exponential distribution that are obtained using predictors from an ensemble of opportunity of climate models.
Again, I don't understand how this assumption would make this number "robust". Robustness is a specific concept in statistics – an estimator is robust if it is insensitive to outliers or misspecification of the underlying distribution. The working assumption (briefly discussed in the supplement), is that the available ensembles of opportunity represent model uncertainty adequately (uncertainty arises from structural and parametric differences, and from internal variability), and that all simulations from all models are equally representative of variations that can be associated with model uncertainty. Trimming the "sample" at the 95th percentile would, under that assumption, add a measure of robustness from a statistical perspective, but this relies on the strong assumption that the available ensemble of opportunity can be conceived of as a random sample of climate change simulations that is representative of a well-defined population of plausible representations of the climate system.

*Some specific comments* (numbering refers to page and line number within page):

1, 24: This statement incorrectly describes the information that is presented in the Munich Re report.  They count loss events that are weather related, not weather events that are loss-relevant. While the difference is subtle, it is important to understand that the focus of the Munich Re report is losses.

2, 7-9: This is changing; I'm aware of at least one large-ensemble experiment similar to the NCAR large ensemble in which an RCM is being used to construct a large ensemble by downscaling a large ensemble of global simulations.

2, 29-31: Why would this be a more tractable question and have greater prospects of overcoming the problems cited in the preceding lines?

3, 15-18: Why use NCEP/NCAR-1 in preference to some other source of North Atlantic surface air temperature or SSTs. How is GCM output bias corrected?

4, 3-5: It's unclear how this 90% "confidence interval" is constructed.  Is it really solved as a parametric problem that involves a distributional assumption, or is this

simply a matter of finding the central 90% range in the ensemble of results obtained by using the available collection of climate simulations? As an aside, I don't think trying to say that the two approaches produce similar results increases "confidence". Also, I would recommend simply referring to an "uncertainty range" rather than a confidence interval, since the statistical basis for calculating a confidence interval seems unclear (see my comments concerning robustness above).

P5, 9-24: It would be useful to start here by giving a complete description for the cumulative distribution function for X, which would include a statement about the probability that X=0, as well as a description of the probability that X>0. Note that standard statistical notation usually reserves lower case letters for probability density functions, and uses upper case letters for cumulative distribution functions.

P6, 15: Why this particular threshold for $R^2$?

P6, 18-19: I'm not convinced that the strategy has achieved the goal that is stated here. With enough caveats, you might convince readers that you have estimated an upper bound for a 90% uncertainty interval, but that's a far cry from evaluating "the maximum potential effect of temperature changes on the wet day mean".

7, 2-5: What is the physical explanation for the spatial variability that is seen in Fig. 3? Can we consider the spatial variation to be "robust", as opposed to being the result of statistical or modelling artefacts?

7, 22-23 "worst case": See previous comments on the communication of what this paper is trying to do.

8, 11-14: What about mid-latitude coastal regions affected by atmospheric rivers?

8, 24: See previous comments about robustness. Furthermore, because a formal statistical framework is not used to perform this work is adhoc, I think it is unclear whether estimates produced from this approach are more efficient (ie, "make the most out of the available information") than competing estimates. In general, robustness and efficiency can be somewhat opposed to each other, although one objective of statistical robustness is to limit losses of efficiency due to a change or misspecification in distribution. Perhaps one could demonstrate "robustness" by showing that there is only modest sensitivity to excluding or adding the "outlier" model according to some measure of model performance vis-à-vis extreme precipitation.

---

## Referee Comment (RC2) · R. V. Donner (Referee) · 24 Sep 2016

Benestad et al. present a methodology for statistical-empirical downscaling of precipitation time series to estimate upper limits for future return levels. The proposed methodology provides a considerable alternative in cases where more explicit estimates are not available. In general, the manuscript is carefully written and accompanied by very detailed supplementary material. In fact, my impression is that in some cases, the reader finds relevant information regarding the motivation and methodological details only in this supplement, and one might discuss if some of the supplementary material might fit better into the main paper.

In general, I recommend publication of this manuscript in NHESS after certain revisions have been made. Below, I provide a list of specific recommendations that the authors
might wish to consider when preparing their final manuscript.

1. I acknowledge that the authors use well-studied data from the CMIP5 ensemble. In the context of the present work dealing with estimating future return levels, it would be advantageous if the authors could briefly summarize some information on potential known biases (if there are any) of the considered projections.

2. Referring to the statement that "heavy precipitation will become more severe in already wet areas in the future " (p.1, ll.25-26), I was wondering if this holds globally in all such regions.

3. The proposed method relies on inferred statistical relationships between different climate variables. A few more words on possible limitations of these relationships (from both physical principles and empirical observations) would be useful.

4. The authors state several times that their estimates provide upper limits. From the presented material, I did not fully understand why this is the case. The argument seems to refer to the relationship between the variables used for empirical-statistical downscaling; however, the results would only be an upper limit if other (unconsidered) covariates would have exclusively opposite effects and could not enhance the considered relationship. Is it possible to rule out (from physical principles) possible "positive interferences" between different variables possibly influencing precipitation?

5. The proposed method is based on an exponential distribution of 24-hours precipitation sums, whereas I would naively expect a gamma distribution being a more common statistical model (even though the simple scaling from the behavior of the mean to that of arbitrary quantiles would not apply anymore in such case). I would be interested in some additional details on why the exponential distribution is justified here.

6. The choice of the reference region in the North Atlantic appears to be motivated by general climatological considerations rather than statistical optimization. Could the results of the empirical-statistical downscaling be further improved by explicitly seeking for the strongest statistical relationships between predictor and predictand fields? Specifically, as the authors recognize, their predictions are not very convincing in regions with complex orography – could this be because the predictors are not appropriately chosen for these locations in terms of their geographical spread? Can the considered relationship be assumed to be essentially homogeneous over entire Europe?

7. In Eq. (1), is the considered noise term white or serially correlated?

8. The PCA in Section 2.4 predetermines mutually orthogonal annual cycle shapes in PC1 and PC2. It is not clear if and why this is desirable in the present case. Specifically, what the authors consider here in terms of the coefficients of PC1 (PC2) is closely related to the phase of the annual cycle, since both components essentially generalize the role of sine and cosine functions in case of a fully harmonic oscillation (PCA is commonly based on normalized time series, so amplitudes do not matter that much ). It might be useful to directly refer to some corresponding phase variable to parameterize the shape of the seasonal cycle for each considered location.

9. In a few figures (in both main manuscript and supplementary material), axis labels and labels/units to color bars are missing. This should be carefully revised. In Fig. 3, it is not clear if the inset shows relative or absolute changes.

Technical comments:

* p.3, l.3: "Our approach. . ." would rather call for using present tense.

* p.4, l.18: mathematical symbol missing after "referred to as"

* p.4, ll.25: "the ration between explained variation. . . and the total variation. . ."

* p.4, l.25: "var() with the noise term is taken to be zero" is not quite understandable, please rephrase

* p.4, l.25: "Principal component analysis"

* p.7, l.13: "constant value of the…"

* p.8, l.1: "dependency… on temperature"

There are also a few typos in the supplementary material that are not listed here for brevity. In general, in some of the R outputs embedded in the SM text, the meaning of the individual variables is not fully clear without consulting the full R code; at least identifying the variables in the corresponding text boxes would facilitate the reading. Also, I did not find a caption for Fig. SM14.

---

## Author Comment (AC1) · 17 Oct 2016

We are grateful for the comment regarding a potential confusion concerning probable maximum precipitation, which is different to the maximum systematic effect that a temperature change may have on the precipitation. The use of an upper limit, however, was inspired from use in physics where problem solving sometimes involves the estimation of upper and lower limits if the most likely estimate is difficult to derive.

The Greek letter $\mu$ was used to represent the wet-day mean precipitation which is also related to the parameter of the exponential distribution f(X)= $\lambda$ exp(-$\lambda$x), where $\lambda$=1/$\mu$. The best fit to this distribution is sought for various data samples, such as on an annual or monthly basis. Hence, there will be different estimates of $\mu$ for different calendar months. This is one of the new aspects of this strategy, and we are grateful for the

reviewer pointing out how this may cause confusion. We will try to explain this more carefully in a revised version.

Another aspect is the relationship between $\mu$ and percentiles; it is easy to show mathematically that for the exponential distribution, any percentile can be written q_p = -ln(1-p) $\mu$ (this expression is derived in http://dx.doi.org/10.3402/tellusa.v67.25954). We think that the use of three different symbols for the same quantity would be more confusing than using one, as the Greek letter $\mu$ refers to the wet-day mean precipitation for all these instances.

We disagree with the comment of the strategy being ad-hoc: with climate change, we must expect a change in the probability distribution function (pdf) and that changes in the tail of the pdf must follow the change in the bulk of the pdf as the area under its curve must equal to one by definition. The use of an exponential distribution does not give a precise representation of the tail of the distribution, but since it only has one parameter, it gives a constrained description of the change in the bulk part of the probabilities.

We agree that GEV would be a better choice for describing extremes for a stationary variable, but it is more questionable if it is the best method to quantify changes in the extremes for a non-stationary situation with short data records. Hence, there is a trade-off between using the less precise but more robust one-parameter pdf that takes changes in the entire pdf into account or the three-parameter GEV that only considers the tail of the distribution for fitting.

We used the exponential distribution to quantify the percentile for highest annual precipitation, which is not so far out in the tail. This percentile is then used together with a 95-percentile (1-in-20 year) of the annual aggregate for the wet-day mean precipitation $\mu$ to estimate the 20-year return value, inspired by Bayes equation. However, we realise that this must be explained more carefully to avoid confusion.

Our method does indeed assume that the wet-day frequency is not changing, and this

is of course debatable. Analysis of past trends suggest that there has been mixed long-term changes (Benestad et al., 2016; ERL-102170.R2 - supporting material).

The upper bound is taken to be the assumption that all of the mean seasonal variability in $\mu$ is due to the seasonal variability in the temperature in parts of the North Atlantic which is a likely source of the atmospheric moisture.

This paper uses both physics and statistics, and there are sometimes different conventions in these two disciplines. We are not pretending that this paper is a pure statistics-study, but try to draw on common practices from statistics as best as possible, but not to the extent that it interferes with common practices in physics. We are also aware of past criticism that statisticians have made about physics and vice versa. We think that our use of "robust" meets both that of statistics (our estimates are not sensitive to outliers) and physics (they rely on a high signal-to-noise ratio since we make use of the mean seasonal cycle). This will be explained more carefully in a revised version.

We are aware of the fact that a multi-model ensemble like CMIP5 is not ideal, but is in reality "an ensemble of opportunity". We also acknowledge that sampling strategy is key to statistical reasoning, however, this is not such a critical point in this work. We do not use the ensemble to assess model-related uncertainty, but we find that the ensemble spread corresponds well with the interannual variability. We choose to adopt a common practice in physics, to be practical and make use of what information that is available. We choose to make use of a rough "back-of-the-envelope" approximation to be able to get a reasonable - although not perfect - estimate of the largest effect a temperature increase may have on extreme precipitation.

We appreciate the specific comments. To be rephrased: number of loss events, related to weather. Can be rephrased to 'has been', but still with a larger ensemble, such as Euro-CORDEX, it is much smaller than for the ensembles used in ESD as done here with hundred simulations for the most popular emission scenarios. This is more a physics way of addressing a problem. NCEP/NCAR 1 is used because it has longer

data records and it has some similarities to a GCM being output from an atmospheric model. The predictor is the area mean temperature and such an aggregated quantity reduces the effect of a bias. The precise description is given in the R-markdown in the supporting material. The ensemble is used to gauge the interannual variability to estimate a typical 1-in-20 year warm annual wet-day mean precipitation $\mu$. We will explain this more carefully. We are not sure what is the problem here. Pr(x >0) = fw, which we expect. Pr(X=0) = 1- fw. The threshold is a bit arbitrary, but it was chosen to highlight the locations with a good match between the mean seasonal variations in $\mu$ and in the temperature over the North Atlantic. This is a matter of debate, and I hope that our paper can be considered as one contribution. We make use of conditional probabilities and the assumption that all of the mean seasonal variations in $\mu$ are due to the mean seasonal variations in the temperature over the North Atlantic. Comparison with independent data give some support for this (supporting material). Good question. The outer rims indicate more geographical variance, and are probably subject to stronger statistical fluctuations connected to the stronger model response. These may depend on local geographical conditions, and since the results are based on rain gauge data, differences may be due to instrumentation and surrounding obstacles. Here, robust refers to the use of a "cleaner" signal compared to noise, in the use of the mean seasonal cycle. It also downplays the effect of outliers in terms of single events. The paper does indeed try to quantify a "worst-case" estimate based on the assumption that all of the mean seasonal cycle in $\mu$ that matches the mean seasonal cycle in the temperature is due to the change in temperature. This needs to be explained more carefully. Atmospheric rivers are phenomena taking place in the upper part of the troposphere. They transport humidity from low latitude regions and cause heavy precipitation at higher latitudes. They may of course be part of the equation here, but we do not see any reason why they should play a special role and we are not aware of a direct effect between these and the coasts. This is something that should be looked into further, but is outside the scope of our paper. This paper is more from a physics approach, and we disagree that anything that is not statistical is "ad-hoc". GEV-based work is pure

**NHESSD**

statistics, but our strategy made use of the information contained in the mean seasonal cycle and an upper-limit estimate based on a physics problem solving strategy that has been part of the physics training at University of Manchester Institute of Science and Technology (UMIST).

———————————————————

---

## Author Comment (AC2) · 17 Oct 2016

1. We are grateful for the comments made by Reik Donner, who rightly points to the issue of biases connected with the GCMs in CMIP5. We made use of area averaged temperatures, and it's a good idea to add some evaluations of the mean bias as well as biases in the spread of interannual variability and past trends. An account on this aspect will be included in a revised version.

2. The concern about whether the statement that "heavy precipitation will become more severe in already wet areas in the future" holds globally in all such regions is indeed valid. We do not have the analysis for the entire globe here, but we can refer to previous studies (e.g. Benestad, R.E (2006) Can we expect more extreme precipitation on the monthly time scale? J.Clim Vol. 19, No. 4, pages 630-637) as well as the IPCC

[Figure]

AR5.

3. There may be other processes that cause a mean seasonal variations in $\mu$, especially if they are correlated with the mean seasonal cycle. Hence, the use of an upper limit. On the other hand, there is physics which explains the connection between temperature and air moisture (Clausius-Clapeyron). We will add some more text on these caveats.

4. It is possible that some other process that also varies with the means seasonal cycle also affects $\mu$ in a way that is indistinguishable from the mean seasonal cycle in the temperature over the North Atlantic. This is one of the main caveat of this strategy. We are not aware about possibilities to rule out positive interferences from physical principles.

5. It is correct that the gamma distribution is more commonly used for 24-hr precipitation, however, it is more difficult to fit and it is less constrained with two parameters. Furthermore, past analysis suggests that the exponential distribution (which is a special case of the gamma distribution) gives an approximate representation of the frequencies for when only wet-days (more than 1 mm/day) are considered. We use the exponential distribution because is gives an approximate fit to the data (Benestad, R.E., D. Nychka and L.O. Mearns, 'Specification of wet-day daily rainfall quantiles from the mean value', Tellus A, 64, 14981, DOI: 10.3402/tellusa.v64i0.14981; Benestad, R.E., D. Nychka and L.O. Mearns Spatially and temporally consistent prediction of heavy precipitation from mean values, Nature climate Change, doi:10.1038/nclimate1497) and the single parameter distribution makes it easier to infer changes in the tail as a consequence of a change in the bulk shape of the pdf. This will be more carefully explained in the revised version.

6. We chose a region according to our understanding of the physics and expectations about it being a source region for moisture. We could probably find a more optimal area, but it would involve a kind of "cherry picking" and we would need to account for multiple

tests through field significance. We also think that different geographical conditions influence the presence of different phenomena all which affect $\mu$. In the interior regions, it is primarily summertime convective events that give high amounts, whereas for the mountain range along the west coast of Norway, it seems to be cyclones and orographic precipitation.

7. The noise term may for all intents and purposes be considered to be white as there is little autocorrelation from year to year.

8. We used PCA to compress the information in the data and to simplify the analysis, as well as to identify canonical shapes in the seasonal cycle. The weights (PCs) were used in the latter to identify locations with similar means seasonal cycle characteristics.

9. Thanks for pointing this out.

10. Thanks for the technical comments - they will be taken into consideration in a revision.

---

## Author Response (AR1)

**Reviewer 1**

I think there is considerable substance in this paper, but confusing presentation and imprecise language gets in the way of making a convincing case for the approach that is proposed. This is apparent even from the title. As soon as one combines the words "upper-limit" with "precipitation" in the same sentence, one evokes implicitly the idea of probable maximum precipitation – ie, an amount that is a physically plausible upper bound. The language in this paper is sufficiently imprecise that confusion about whether the aim is to estimate an upper bound for precipitation arises at several places. This is reinforced even within the abstract, where we read that the proposed method "utilises ... to estimate the maximum effect that temperature change can have on precipitation ...". Since the paper is about extreme precipitation, and the title refers to "upper-limit estimation", one could be excused for thinking that the interest is in estimating absolute upper bounds for 24-hour precipitation accumulations.

The purpose of the paper is to estimate future return-values in circumstances when there are limited observations, where traditional methods are not applicable. The alternative that we present is calibrated on larger sample sizes (the mean climatology) stretching over longer time periods, which puts more weight on slow processes with long time scales. It is an estimate of the upper limit of the influence of temperature on precipitation in the sense that we assume that the seasonal cycle of the wet-day mean can be explained solely by variations in the temperature in the predictor domain. However, as one of the reviewers pointed out, other factors may also add to a precipitation increase, so it is a bit misleading to call it an upper bout. We use the term potential sensitivity to draw on analogous concepts such as climate sensitivity but include potential, since this assumes that all of the seasonal precipitation variations are related to seasonal temperature variations.

Awkward mathematical notation and poor conversion of the typescript to a pdf document also contribute confusion. At various places the reader has to infill his/her own guess of what symbols are meant to be present in the text, apparently because they simply have disappeared, or have been misplaced, in the process that rendered the review document. More importantly, symbols such as the Greek letter  $\mu$  seem to be used in multiple ways, with the reader needing to infer the correct interpretation from the context in which the symbol appears. For example,  $\mu$  is used as the parameter of a probability distribution, as a quantity that varies from month to month, and as a quantile. It would be wonderful if standard statistical conventions could be used for notation. Generally, this means using Greek letters to represent unknown parameters that are to be estimated from observations, Greek letters with hats to denote parameter estimates, upper case Latin characters to represent random variables, and lower case Latin characters to represent realizations of those random

variables (either observed or simulated), etc.).

We used Overleaf and LaTeX for our revised version which has a better handling of mathematical symbols and equations. The greek letter  $\mu$  is commonly used for the wet-day mean precipitation and  $f_w$  for wet-day frequency, and we would like to keep it that way. However, we have tried to make it easier to differentiate between observations and downscaled values by being more consistent in the use of hats to represent values estimated with the downscaling downscaling models.

The approach itself seems rather ad hoc. The fundamental working assumptions are apparently that

a) an exponential distribution fitted to all non-zero daily precipitation amounts will nevertheless represent most of the upper tail of the precipitation distribution reasonably well, including as far into the tail as 1-in-20-year 1- day events, and that b) precipitation frequency will not change. The first, (a), is actually a very strong assumption given the existing body of observational evidence from multiple sources that suggests that extreme daily precipitation is heavy-tailed (that is, Frechet distributed rather than Gumbel distributed). A consequence of assuming the exponential distribution for daily precipitation is that block maxima (such as annual maxima) will converge to the Gumbel distribution (GEV distribution with shape parameter equal to zero) rather than the Frechet distribution. The second assumption also seems a strong assumption given the sensitivity of the interpretation of changes in quantiles of non-zero precipitation amounts to changes in frequency (e.g., see Schar er al, 2016, doi: 10.1007/s10584-016-1669-2). To be fair, the authors defend (b) in this paper, and have discussed (a) in previous papers. Nevertheless, these are strong assumptions that work against the claim that the method provides robust estimates of an upper (uncertainty) bound for 20-year return values.

The point here with combining the 1-in-20 year events and annual maximum daily values is that we do not extend far into the tails of the distribution and in terms of those samples, we look at moderate extremes. This is different to the traditional methods where the outer parts of the tails are modelled. The point about the frequency, on the other hand, is a genuine caveat that we discuss in the paper. There is also additional discussion of this issue in the SM.

A third key assumption is that it is assumed that

c) to the extent that 20-year return values are sensitive to temperature, uncertainty in projected changes in 20-year return values can be bounded by using the 95th percentile of changes in the parameter of the exponential distribution that are obtained using predictors from an ensemble of opportunity of climate models. Again, I don't understand how this assumption would make this number "robust".

Robustness is a specific concept in statistics – an estimator is robust if it is insensitive to outliers or misspecification of the underlying distribution. The working assumption (briefly discussed in the supplement), is that the available ensembles of opportunity represent model uncertainty adequately (uncertainty arises from structural and parametric differences, and from internal variability), and that all simulations from all models are equally representative of variations that can be associated with model uncertainty. Trimming the "sample" at the 95th percentile would, under that assumption, add a measure of robustness from a statistical perspective, but this relies on the strong assumption that the available ensemble of opportunity can be conceived of as a random sample of climate change simulations that is representative of a well-defined population of plausible representations of the climate system.

Well, as the reviewer pointed out, we do discuss assumptions (a) in the paper and (b) in several previous papers. The robustness lies in the fact that we use a much larger sample size when calculating the mean climatologies that the downscaling model of mu is based on. We have changed the text to specify that it is "robust to outliers" (because of the larger sample size). It is true that the method is not robust to misspecifications (but no method ever is that) of the underlying distribution, but we do discuss the assumption of an exponential distribution.

As for point ( c ), our use of the ensemble was to represent local and regional variability of the climate system, which is strongly affected by internal variability. We discuss this in the supporting material, but then say "Nevertheless, the spread of downscaled annual mean temperature from ensemble experiments such as CMIP5 is often comparable to the magnitude of the observed year-to-year temperature variations..." and go ahead and use it as such anyways. We have added a sentence of caution in the main manuscript.

Some specific comments (numbering refers to page and line number within page):

1, 24: This statement incorrectly describes the information that is presented in the Munich Re report. They count loss events that are weather related, not weather events that are loss-relevant. While the difference is subtle, it is important to understand that the focus of the Munich Re report is losses.

**The sentence has been changed to better reflect the content of the Munich Re report.**

2, 7-9: This is changing; I'm aware of at least one large-ensemble experiment similar to the NCAR large ensemble in which an RCM is being used to construct a large ensemble by downscaling a large ensemble of global simulations.

This may be changing - there may be cases where RCMs are applied to larger ensembles of GCMs - but computational demands are still a limitation and certainly have been in the past. The text has been changed a little so that it refers more to past studies and doesn't exclude all possibility of ever applying RMCs to large ensembles.

2, 29-31: Why would this be a more tractable question and have greater prospects of overcoming the problems cited in the preceding lines?

**The text has been changed here.**

3, 15-18: Why use NCEP/NCAR-1 in preference to some other source of North Atlantic surface air temperature or SSTs. How is GCM output bias corrected?

**This reanalysis extends back to 1948 and the surface air temperature is more comparable to output from GCMS in the CMIP experiment than observations-based SSTs. GCM output bias was not corrected and we used spatially aggregated estimates of $e_s$ over a large region (100W-30E/0-40N) as predictors.**

4, 3-5: It's unclear how this 90% "confidence interval" is constructed. Is it really solved as a parametric problem that involves a distributional assumption, or is this simply a matter of finding the central 90% range in the ensemble of results obtained by using the available collection of climate simulations? As an aside, I don't think trying to say that the two approaches produce similar results increases "confidence". Also, I would recommend simply referring to an "uncertainty range" rather than a confidence interval, since the statistical basis for calculating a confidence interval seems unclear (see my comments concerning robustness above).

**It is simply a matter of finding the central 90% range in the ensemble of results obtained by using the available collection of climate simulations. We changed the term confidence interval to uncertainty range.**

P5, 9-24: It would be useful to start here by giving a complete description for the cumulative distribution function for X, which would include a statement about the probability that X=0, as well as a description of the probability that X>0. Note that standard statistical notation usually reserves lower case letters for probability density functions, and uses upper case letters for cumulative distribution functions.

**The case for X=0 is trivial and is accounted for by the wet-day frequency $1-f_w$ . We change the notation 'f(.)' to 'Pr(.)'**

P6, 15: Why this particular threshold for R2?

**There has been a tradition that R2 needs to be at least 0.6 for practical use in terms of skillful forecasts. This is of course subjective.**

P6, 18-19: I'm not convinced that the strategy has achieved the goal that is stated here. With enough caveats, you might convince readers that you have estimated an

upper bound for a 90% uncertainty interval, but that's a far cry from evaluating "the maximum potential effect of temperature changes on the wet day mean".

**The description of the estimate has been changed somewhat to emphasise that it is an approximate estimate of future return-values, and that the main advantage is that it is applicable in cases of limited data availability.**

7, 2-5: What is the physical explanation for the spatial variability that is seen in Fig. 3? Can we consider the spatial variation to be "robust", as opposed to being the result of statistical or modelling artefacts?

The most obvious physical explanation of the spatial pattern is orographic effects as it is limited to the Alpine region and western Norway. The PCA analysis of the seasonal cycle of observed wet-day mean precipitation also pointed to a different precipitation regime in these areas. Since there is an obvious physical explanation and the PCA analysis and regression analysis both pointed to the same geographical pattern, we see no reason to suspect statistical or modeling artefacts.

7, 22-23 "worst case": See previous comments on the communication of what this paper is trying to do.

**We changed the wording and removed the term "worst-case".**

8, 11-14: What about mid-latitude coastal regions affected by atmospheric rivers?

The atmospheric rivers may be excluded here, but that depends on their frequency and the character of their seasonal appearance. The weights of the PCA for the seasonal cycle are more typical of convective events here. The atmospheric rivers and convective events represent different phenomena and one should not expect to have one statistical framework that fits all such.

8, 24: See previous comments about robustness. Furthermore, because a formal statistical framework is not used to perform this work is adhoc, I think it is unclear whether estimates produced from this approach are more efficient (ie, "make the most out of the available information") than competing estimates. In general, robustness and efficiency can be somewhat opposed to each other, although one objective of statistical robustness is to limit losses of efficiency due to a change or misspecification in distribution. Perhaps one could demonstrate "robustness" by showing that there is only modest sensitivity to excluding or adding the "outlier" model according to some measure of model performance vis-à-vis extreme precipitation.

This approach is a hybrid between a physics problem-solving approach and statistical thinking, and therefore will appear as ad hoc to the pure physicist or statistician. The point is the practicality and our approach is more computationally efficient than many other methods. This can be seen because it is applied to a large ensemble of models. The calculations take a very short time, and the study includes a number of validation exercises to evaluate its merit.

**Reviewer 2**

Benestad et al. present a methodology for statistical-empirical downscaling of precipitation time series to estimate upper limits for future return levels. The proposed methodology provides a considerable alternative in cases where more explicit estimates are not available. In general, the manuscript is carefully written and accompanied by very detailed supplementary material. In fact, my impression is that in some cases, the reader finds relevant information regarding the motivation and methodological details only in this supplement, and one might discuss if some of the supplementary material might fit better into the main paper. In general, I recommend publication of this manuscript in NHESS after certain revisions have been made. Below, I provide a list of specific recommendations that the authors might wish to consider when preparing their final manuscript.

Some of the relevant discussions from the supplementary material have been included in the main text, but not the supplementary figures.

1. I acknowledge that the authors use well-studied data from the CMIP5 ensemble. In the context of the present work dealing with estimating future return levels, it would be advantageous if the authors could briefly summarize some information on potential known biases (if there are any) of the considered projections. *There is a number of biases, but we used aggregated results in time and space, and found that this then had little effect.*

2. Referring to the statement that "heavy precipitation will become more severe in

already wet areas in the future " (p.1, II.25-26), I was wondering if this holds globally in all such regions.

**We have not checked other parts of the world, as observations are sparse and missing. It is reasonable to infer that this may nevertheless be true if our selection can be considered a random sample from the planet that is representative of the entire system. Convective processes are more or less a universal process on Earth's continents, but may be different over the oceans.**

3. The proposed method relies on inferred statistical relationships between different climate variables. A few more words on possible limitations of these relationships (from both physical principles and empirical observations) would be useful.

**We refer to the scaling between the two as 'potential sensitivity' exactly to communicate potential limitations.**

4. The authors state several times that their estimates provide upper limits. From the presented material, I did not fully understand why this is the case. The argument seems to refer to the relationship between the variables used for empirical-statistical downscaling; however, the results would only be an upper limit if other (unconsidered) covariates would have exclusively opposite effects and could not enhance the considered relationship. Is it possible to rule out (from physical principles) possible "positive interferences" between different variables possibly influencing precipitation?

**Good point. The text has been changed somewhat because the term upper limit can be misleading, and we use the term potential sensitivity. However, it still is an upper limit, but we try to explain this more carefully.**

5. The proposed method is based on an exponential distribution of 24-hours precipitation sums, whereas I would naively expect a gamma distribution being a more common statistical model (even though the simple scaling from the behavior of the mean to that of arbitrary quantiles would not apply anymore in such case). I would be interested in some additional details on why the exponential distribution is justified here.

**The exponential distribution as a description of precipitation is discussed in several previous papers, referenced in the paper. We used the exponential distribution for simplicity as it requires the estimation of just one parameter which is the mean.**

6. The choice of the reference region in the North Atlantic appears to be motivated by general climatological considerations rather than statistical optimization. Could the results of the empirical-statistical downscaling be further improved by explicitly seeking for the strongest statistical relationships between predictor and predictand fields? Specifically, as the authors recognize, their predictions are not very convincing in regions with complex orography – could this be because the predictors are not appropriately chosen for these locations in terms of their geographical spread? Can the considered relationship be assumed to be essentially homogeneous over entire Europe? Some tests of predictor domain were conducted on a few test sites, but no systematic optimization. The fact that the predictions are not useful in complex terrain is most likely due to different processes influencing the orographic precipitation (atmospheric circulation etc) than convective precipitation (temperature and moisture). (The same spatial pattern was seen also in the PCA of the seasonal cycle of the wet-day mean.) The predictor domain should be adapted to the predictand to reflect the main moisture source, but this domain should work ok in other parts of Europe as well.

7. In Eq. (1), is the considered noise term white or serially correlated? *White*

8. The PCA in Section 2.4 predetermines mutually orthogonal annual cycle shapes in PC1 and PC2. It is not clear if and why this is desirable in the present case. Specifically, what the authors consider here in terms of the coefficients of PC1 (PC2) is closely related to the phase of the annual cycle, since both components essentially generalize the role of sine and cosine functions in case of a fully harmonic oscillation (PCA is commonly based on normalized time series, so amplitudes do not matter that much ). It might be useful to directly refer to some corresponding phase variable to parameterize the shape of the seasonal cycle for each considered location.

The PCA is simple and not constrained by the shape (like sinusoids), and we wanted to identify the covariance structure in the mean seasonal variations. The orthogonality is nice when using them in regression analysis, but of course, the variations themselves are a superposition of several modes.

9. In a few figures (in both main manuscript and supplementary material), axis labels and labels/units to color bars are missing. This should be carefully revised. In Fig. 3, it is not clear if the inset shows relative or absolute changes.

**We have improved the figures.**

Technical comments:

\* p.3, I.3: "Our approach..." would rather call for using present tense.

**The sentence has been changed/removed.**

\* p.4, l.18: mathematical symbol missing after "referred to as"

A lot of mathematical symbols were missing because of a failed conversion to pdf. The revised manuscript is written in LaTex and all the symbols should be correct.\* p.4, II.25: "the ration between explained variation... and the total variation..."

**This sentence has been removed.**

\* p.4, l.25: "var() with the noise term is taken to be zero" is not quite understandable, please rephrase

This sentence has been removed and the R2 calculations are now explained (and hopefully more understandable) when the results are mentioned later in the manuscript.

\* p.4, I.25: "Principal component analysis"

- \* p.7, I.13: "constant value of the ... "
- \* p.8, l.1: "dependency... on temperature"

There are also a few typos in the supplementary material that are not listed here for brevity. In general, in some of the R outputs embedded in the SM text, the meaning of the individual variables is not fully clear without consulting the full R code; at least identifying the variables in the corresponding text boxes would facilitate the reading. Also, I did not find a caption for Fig. SM14.

We have rearranged and rewritten parts of the text to be more easy to follow. Some of the supplementary figures that were not explicitly referenced in the main or supplementary text have been removed. The R-scripts included has been updated and some more explanation added so that it should be easier to follow it.

---

## Author Response (AR2)

Dear editor,

Thanks for the opportunity to address the new round of reviews. Our response is detailed in the the account given below, which is followed by marked-up versions of the manuscript with tracked changes - both the paper and the supporting material. I hope this is OK.

Yours sincerely

Rasmus Benestad

**Response to the reviewer: NHESS**

Report #1

In the response letter (reply to referee 1), the authors agree that "it is a bit misleading to call it [the obtained type of return-level estimate] an upper bound" and state that they use the term potential sensitivity instead. However, the terms "upper bound" and "upper limit" still occur at various places, particularly the abstract (l.9), method description (ll.90,92), conclusions (ll.285,290) and supplementary material (l.79). While I understand that it might be unavoidable to use such terms in some cases, it is still not fully clear from the reasoning presented in the manuscript why this "potential sensitivity" (of precipitation to temperature changes) is actually an upper bound for the expected precipitation changes and might not be further magnified by other (temperature-unrelated) effects. Clarifying this in the manuscript (with probably just a few more words) still appears necessary.

*Thanks for pointing this out. It is useful to learn that parts of our description was not sufficiently clear, and we have inserted some more text to clarify the issue about "upper bound" and "potential sensitivity". The changes are visible through tracked changes.*

The order or figures in the Supplementary Material is not fully clear to me. If this order shall represent the order of appearance in the main manuscript (as I suppose) the correct order should be SM2, SM3, SM1, SM4, etc. Moreover, if I am not fully mistaken, the R scripts used for obtaining the presented results had been provided in the Supplementary Material of the discussion paper, but not in that of the final paper; thus, the corresponding statement and referencing in ll.100-101 should be corrected.

*Thanks for the suggestions. The SM figures have been rearranged according to the proposed order. The R-code is still part of the supporting material as supporting material, but we forgot to include them in the revised version. They will definitely be included in the final version.*

As has also been criticized in the discussion paper, the variable mu is used in different ways, i.e., for the annual wet-day mean precipitation (l.142) and the observed monthly mean (l.150), which is quite confusing. I suppose that the statistical exponential model of the wet-day mean precipitation as considered by the authors has firstly been developed on an annual basis, but is later being used separately for each calendar month. If this is correct, it would be helpful if the authors could state this explicitly within Section 2.2.

*Thanks for this comment. It was the mean seasonal variation in $e_s$ that was used to train the model for mu, and some text has been added to clearify that.*

In ll.144-145, the authors state that they assume that "factors other than temperature that are affecting wet-day precipitation are stochastic and stationary". However, the validity of the stationarity assumption seems to be tested only for the wet-day frequency (Section 1 of the Supporting Material). A few words on this aspect at the mentioned position in the text would be helpful.

We have added a couple of lines about interdependencies and our heuristic physical picture of the atmosphere.

Several references appear incomplete, especially Benestad et al. 2012a (pages missing, also in SM), Berg et al. 2013 (volume and pages missing, typo in journal name), Takayabu et al. (volume and pages missing) and Benestad 2008 (only SM – what is this reference?).

*These details have now been included.*

In general, the validity of the exponential approximation of the PDF of the wet-day precipitation might have been shown in the existing literature, but the quality of this approximation should still be discussed briefly in the manuscript or at least in the SM. Specifically, I understand the authors' argument that they consider only moderate extremes (1 in 20 years, 95% quantiles, cf. p.2 of the response letter to the original reviews) – but then, they should state this explicitly in the text and emphasize that the approach is likely to perform less well if even rarer extremes are considered (say, 50 or 100-year return levels).

*This has now been added*.

In connection with Fig. SM9, it would be good if the authors could highlight the stations with statistically significant trends (e.g., by a black circumference of the filled circles). In the caption of Fig. SM8, it is mentioned that there are regions with significant trends; these are, however, hard to assess in Fig. SM9. It should also be briefly stated how significance should be assessed here (i.e. if the statistical independence assumption of a classical t-test would hold or if there are sufficiently strong serial correlations in the historical records that would call for more complex testing approaches like block bootstrapping).

*In both Fig SM7 and SM9 the locations with significant trend (5%-level) have been highlighted. The test was based on a regression analysis and the p-value associated with the fitted slope of a linear fit.*

In relation with Fig. SM10, please add a very short explicit statement on the selection criteria for the chosen stations. Was it just time series length and data quality? This is just to clarify that they might not be any selection bias.

*Only stations with more than 20000 valid data points were selected, and only the 1945--2015 period was used.*

% Technical comments:
% -      Line 24: MunichRe is a reinsurance, not an insurance company. *OK*
*% -      Line 41: "demands have limited". Thanks!*
% -      Data and Methods: Since the introductory paragraphs of this section set the stage for the details presented in the following subsections, use of past tense is rather unusual here.

Please consider using present tense here instead. We use past tense to describe what we did and present tense for valid conditions irrespective of time.

% -     Line 122: remove "in the SM". *OK.*
% -     Line 255: Table 4 should be Table 2. *OK*
% -     Line 281: better start the conclusions in present perfect tense, i.e., "we have proposed…". *OK*
% -     SM, line 1: "additional analyses that address some…". *OK.*
% -     SM, line 14: "influence". *OK.*
% -     SM, line 20: "not too sensitive". *OK*
% -     SM, line 26: "increase over southern Norway". *OK.*
% -     SM, line 28: "typically by the order of 0.1 mm/day…". *OK.*
% -     SM, line 41: "consistent with a near-constant…". *OK.*
% -     SM, line 50: "suggest the highest". *OK.*
% -     SM, line 57: it is not clear what "its density" refers to in this sentence. *The vapour density.*
% -     SM, line 59: "wet-day mean precipitation than temperature…". *OK.*
% -     SM, line 90: "with predominantly orographic". *OK.*
% -     SM, line 102: "South America". *OK.*
% -     SM, line 125: "is often comparable…" – if this is really often the case, one or two corresponding references would be reasonable. *OK.*
% -     Fig. SM1, caption: please add a statement that the results refer to a single selected station only. *OK.*
% -     Fig. SM2, caption: please use \ln to suppress italics in the presentation of the logarithm functions. Ok.
% -     Fig. SM4, caption: "the relative change in comparison to…". *OK.*
% -     Fig. SM5, caption: "between the seasonal cycles in…". *OK.*
% -     Fig. SM6, caption: "long-term linear trends". *OK.*
% -     Fig. SM10, caption: "The inset shows…". *OK.*

**Report #2.**

The authors present an improved version of their previous submission. However, there are still parts that are unclear and there is some poor writing in places.

**We have re-read and revised the paper.**

The conclusions section discusses the assumptions and the key outcome of the analysis. What is missing are some statements regarding the scope of validity of the study. Which assumptions have been shown to be consistent with the historical data? How robust is the result given the limited number of RCMs used and the focus on a single scenario?

*This paper only mentions RCMs in the introduction, but the results were not based on RCM simulations. The results presented here involve several emission scenarios (RCP 2.6, 4.5, and 8.5) in addition to multi-model ensembles. The analysis was in large parts based on historical data. The comment does not make sense.*

% The other reviewer also highlights some poor writing. Examples of poor writing are:
%
% P.2 - "The use of RCMs ARE", should be is.
% P.2 - "and phenomena but subject". Please add comma before but.

%
% Please have a careful read through your document for such issues.
%
% Further, there are some unclear sentences. For example:
%
% P.2 - "...associated with uncertainties from a number of sources, many of which are connected with methods..." What are the methods you are referring to here?
%

*I cannot find these passages in our manuscript. These comments do not make sense.*

[revised manuscript text omitted]

[c1] It
[c2] *Text added.*
[c3] an upper limit
[c4] *Text added.*
[c5] *Text added.*

95 temperature may inflate the role of the temperature, as other factors exhibit a similar mean seasonal cycle and may have an influence on the precipitation intensity. For this reason, we use the terms "upper bound" and "potential sensitivity". It is also true that other unaccounted-for processes possibly may influence the precipitation intensity in a nonlinear fashion and possibly result in even higher intensities if they also change in the future. However, as long as (a) such factors have an approximately

100 linear dependency on the temperature and (b) the temperature may be taken as a proxy for climate change, then this simple assumption may provide a reasonable figure. [c6]This simple method differs from traditional methods in that rather than attempting to specify the *most likely* value, it estimates a kind of *upper [c7]bound* of the systematic response of extreme precipitation to changes in temperature. We henceforth describe this relation as the *potential sensitivity* (PS) since the calibration used the

[revised manuscript text omitted]
.** This supporting material provides additional [c1]analyses that addresses some of the assumptions made in the main paper. It also explains the strategy that we chose and to emphasise this has been structured as questions and answers. The analysis presented here was carried out with the open source R-package 'esd' (Benestad et al., 2015). An R-markdown script with the step-by-step
5    code of the analysis is available from figshare.com for the sake of traceability and replicability (DOI: 10.6084/m9.figshare.4476419)

[c1]

**1   Is the wet-day frequency stationary?**

In this paper, we estimate future return-values of precipitation based on temperature projections, but neglect to evaluate changes in the wet-day frequency ($f_w$) and simply assume it to be stationary.
10   How does this assumption hold up? Has the wet-day frequency varied significantly in the past and do we expect large changes in the future? To answer these questions we have studied the seasonal cycle and past trends of the wet-day frequency.

    Changes in the wet-day frequency affect the probability for heavy precipitation amounts in the future according to $Pr(X > x) = f_w \; e^{-x/\mu}$, and hence influence[c2] future return-values according
15   to $x_{1yr} = \mu \ln(365.25 \times f_w)$. This goes for both long-term changes (trends) as well as interannual to decadal variations. Historical precipitation observations can be used to estimate the interannual variability of $f_w$ and its effect on $x_{1yr}$, but short records mean that the sample size is limited and may preclude a complete account of the effect of decadal changes.

[c2] s

    The wet-day frequency responds weakly to the seasonally varying conditions (Figure SM3; grey
20   curve), which suggests that it is not [c3]too sensitive to systematic changes in the state of the local environment. We can also make use of some information from past trends in the wet-day frequency, as

[c3]

climate change is already happening (Figures SM8 and SM9). Historical data suggest different tendencies in different regions (Figure SM9), and previous analysis indicates that the wet-day frequency is strongly influenced by the circulation patterns (Benestad and Mezghani, 2015). The analysis of

25    historical precipitation records over the period 1961-2014 show little trend in $f_w$ when taking the mean over all locations (Figure SM8), and the only clear spatial pattern is an increase [c4]over south-    [c4] *of* ern Norway (Figure SM9). This should be compared to the wet-day mean precipitation $\mu$ which for most of the sites increased during the same period, typically [c5]by the order of 0.1 mm/day per    [c5] *Text added.* decade (Figures SM6 and SM7).

30    **2   Does variation in the wet-day mean precipitation really correspond to changing probabilities?**

The probability framework adopted here can be formulated as $Pr(X < x|\mu)$, meaning that it is conditional on the sample mean of $\mu$ and that the distribution is exponential. Previous studies have found that the wet-day daily precipitation is approximately exponentially distributed (Benestad and

35    Mezghani, 2015; Benestad et al., 2012b; Benestad, 2007; Benestad et al., 2012a), albeit with a systematic bias connected to the location. The assumption can be assessed by comparing the actual percentiles with quantiles estimated for different samples with different annual mean $\mu$ using the formula for exponentially distributed data:

$$q_p = -ln(1-p)\mu. \tag{SM1}$$

40        The exponential distribution implies a similar proportional change for all percentiles, which is roughly consistent [c1]with a near-constant ratio of increase in daily precipitation percentiles above the    [c1] *Text added.* $90^{th}$ percentage (Pall et al., 2007). The two quantities should be similar (as Figure SM1 indicates) and the data scattered along the diagonal in a scatter plot, indicating that a high percentile associated with a low wet-day mean $\mu$ is consistent with a more moderate percentile for a sample with a higher

45    wet-day mean value.

**3   Why use the $100°W - 30°E/0°N - 40°N$ region of the North Atlantic as predictor?**

The choice of predictor region (Figure SM2) in this study was motivated by the idea that the North Atlantic ocean is an important moisture source for precipitation over Europe and prevailing winds suggest that the moisture is transported from the west. Also, the sea surface temperature is highest at

50    low latitudes, which suggest [c2]the highest evaporation closer to the equator. The analysis presented    [c2] *Text added.* here suggests a good match between the seasonal variations of the temperature averaged over this region and the local wet-day mean (see Figure 1 of the main manuscript). The predictor was defined as the area mean saturation vapour pressure and the domain was set after some trials for a few stations, but this crude trial did not involve any systematic study nor any type of fitting/tuning.

**4 Why use the saturation vapour pressure as predictor and not the temperature?**

It is often wise to make use of terms with similar physical dimensions when calibrating statistical models (Benestad et al., 2008). The saturation vapour pressure is proportional to [c1]the vapour density (ideal gas law: $e_s = \rho R_s T$), and the total mass is the product between volume and density. The saturation vapour pressure is expected to be more linearly related to the wet-day mean [c2]precipitation than temperature because their physical dimensions both involve a measure of the water mass. If temperature was used, on the other hand, then the relationship would be expected to be nonlinear due to the Clausius-Clapeyron equation ($e_s = 10^{(11.40 - 2353/T)}$ where $T$ is the temperature in Kelvin).

[c1]

[c2] *Text added.*

**How representative is the exponential distribution for the probabilities associated with heavy precipitation?** The exponential distribution is a simple form for the gamma distribution and has only one parameter $\mu$ determining its shape as opposed two (location and scale) which gives more freedom in the data-fit. None of these, however, are normally used for the estimation of return-periods and general extreme value (GEV) or generalized Pareto distributions usually used to fit the upper tail of the distribution for stationary data where the shape of the PDF does not change. In the non-stationary case, the small sample represented by the upper tail may not provide the best information in terms of the calibration of a changing PDF over time. Since the area under the curve is always unity (probabilities always add up to one), the upper tail is constrained by the rest of the PDF. An approximate way to tackle the changes is therefore to make use of the bulk of the PDF (Benestad and Mezghani, 2015).

**5 Why use the seasonal cycle for model calibration?**

Precipitation is generated by different atmospheric processes and depends on many factors. Hence the signal-to-noise ratio is often low for traditional model calibration based on chronological matching between the amount and some large scale variable such as regional temperature.

One technique commonly used in physics and electronics for optimising the information from systems and measurements with low signal-to-noise ratio involves cycles with well-established frequencies (eg. FM in radio, phase-locking), and in meteorology/climatology seasonal variations is the most pronounced cycle. There has also been some analysis of tropical cyclone frequencies based on the seasonal variations (Benestad, 2009), but there is an important caveat associated with such studies: the seasonal variations in the local insolation may affect both the large scale conditions and the local variable under investigation, and their correlation may reflect the common dependency on this forcing rather than common link. Thus, the assumption that the seasonal cycle in the temperature over the North Atlantic is linked with the seasonal precipitation statistics is the weakest point of this study if one interprets the results as the most likely estimate of the wet-day mean precipitation. Nevertheless, from a physics perspective, it is expected that higher temperatures result in higher evaporation and higher humidity, hence, an increased capacity for greater rainfall amounts.

90 We use the link between the seasonal cycles of $\mu$ and $e_s$ to estimate an upper limit of the effect of a change in temperature on the precipitation, rather than the most likely estimate of the wet-day mean precipitation itself. Calculating a climatological seasonal cycle gives a larger sample size compared to analyses applied on individual years, and gives a value that is based on a sample stretching over longer time periods. Calibration on larger sample sizes stretching over longer time periods puts more

95 weight on slow processes with long time scales.

The link between the seasonal cycles of local $\mu$ and the mean $e_s$ over the predictor domain (Figure SM2) was first assessed by the $R^2$ of the regression. Figure SM5 shows a histogram of the $R^2$ scores, most of which have an explained variance of over 60%. The majority of the stations with poor fits are found in the mountainous parts of western Norway and the Alps (the size of the markers in

100 Figure 3 of the main manuscript are proportional to $R^2$), which indicates that the method proposed here does not work in regions with [c1]predominantly orographic precipitation.                    [c1] *Text added.*

A second level of validation was to compare trends of historical observations of $\mu$ to predicted trends of $\hat{\mu}$ (the seasonal cycle downscaling model applied to the annual mean $e_s$ calculated from NCEP reanalysis temperature data). Figure SM6 shows that there is a more pronounced scatter in

105 the observed trends than the predicted trends, which indicates that factors other than the sea surface temperature, that are not captured by the climatological downscaling model, also have influenced the long-term changes.

The link between the wet-day mean precipitation and temperature is also assessed by extending the analysis to the spatial as well as the temporal dimension. The fact that this relationship exists

110 in two different dimensions is a stronger indicator of a physical link than if it were to be limited to only one. Figure SM10 shows a scatter plot between $e_s$ and $\mu$ calculated based on the local mean daily maximum temperature and precipitation, respectively. The fitted line shows the regression between the local seasonal cycles of $\mu$ and the temperature for 1420 locations (CLARIS data) in South [c2]America, Europe (stations selected for the COST-VALUE experiment 1), and the US (GDCN).     [c2] a

115 The analysis indicates that the wet-day mean (y-axis) increases by 0.4 mm/day per degree C (x-axis) increase of the local temperature if the elevation is accounted for. The coefficient of the spatial regression is generally consistent with the coefficients from the regressions based on the seasonal cycles, within the range of estimated error margins (Figure SM11). An exception was seen in stations located in western Norway and south of the Alps, where the seasonal cycle regression also showed

120 a weak relationship between $\mu$ and $e_s$. It is not expected that the results should be identical, as the climatological temperature involves the mean of the local daily maximum temperature from the stations, whereas the seasonal temperatures were taken from a large region of the ocean and represented daily mean temperature. Nevertheless, similar values for the regression coefficients between $e_s$ and $\mu$ supports the hypothesis that the precipitation amounts are linked to temperature in a way that gives

125 similar changes through the seasonal variations as in spatial variations.

**6 Is the model ensemble spread a good proxy for probabilities?**

Model ensembles do not really provide estimates of probabilities because they cannot be considered as a random sample of data and because they do not give a perfect reproduction of the observed quantities. According to the IPCC "Ensemble members may not represent estimates of the climate

130 system behaviour (trajectory) entirely independent of one another. This is likely true of members that simply represent different versions of the same model or use the same initial conditions. But even different models may share components and choices of parameterisations of processes and may have been calibrated using the same data sets. There is currently no 'best practice' approach to the characterization and combination of inter-dependent ensemble members, in fact there is no straight

135 forward or unique way to characterize model dependence" (Knutti et al., 2010). Nevertheless, the spread of downscaled annual mean temperature from ensemble experiments such as CMIP5 is often comparable to the magnitude of the observed year-to-year temperature variations(Benestad et al., 2016), and the 95th percentile has been used as an approximate estimate of a one-in-twenty year hot summer season (Benestad, 2011). For all intents and purposes, we have used the interval of the

140 ensembles (see, e.g., Figure SM4) as a measure of the variations of the climate system (Deser et al., 2012).

**Test: exponential distribution & changing mean**

[Figure]

Figure SM1

**Figure SM1.** Test for assessing the consistency between the percentiles taken from observations and estimated values using $q_p = -\ln(1-p)\,\mu$ where the values of $q_p$ are estimated using different values of $p$ to compensate for variations in annual mean $\mu$. A critical threshold $x$ can correspond to different percentiles according to $x = q_{p1} = -\ln(1-p1)\,\mu_1 = q_{p2} = -\ln(1-p2)\,\mu_2$.

Figure SM2

[Figure]

**Figure SM2.** The mean air temperature at 2m of the NCEP reanalysis data set over the chosen predictor domain $100^\circ W$-$30^\circ E$/$0^\circ N$-$40^\circ N$.

[Figure]

Figure SM3

**Figure SM3.** A comparison between the seasonal cycle in the mean precipitation, the wet-day mean precipitation, the wet-day frequency, as well as the wet and dry spell lengths for a single selected station. The most pronounced seasonal variations tends to be associated with the wet-day mean rather than the mean precipitation or the wet-day frequency.

[Figure]

**Figure SM4.** An example of projected annual wet-day mean precipitation $\mu$ for the three different emission scenarios RCP 4.5 (grey), RCP2.6 (green) and RCP8.5 (red), expressed as the relative change in comparison to the 2010 values (see Table 1).

[Figure]

Figure SM5

**Figure SM5.** The statistics of the $R^2$ from the regression between the seasonal cycles in the the local wet-day mean $\mu$ and the regionally averaged saturation vapour pressure $e_s$, estimated from the temperature over the seasonal cycles of the surface temperature over the North Atlantic domain ($100°W$-$30°E$/$0°N$-$40°N$; Figure SM2). There is a portion of stations with very low $R^2$ scores, but most stations suggest an explained variance exceeding $60\%$.

[Figure]

Figure SM6

Mean correlation for local year–to–year variations over t=[1961,2014] is 0.2 (−0.04, 0.41)

**Figure SM6.** A comparison between the long-term linear trends estimated from the observed annual mean $\mu$ and $\hat{\mu}$ values estimated with Equation 1 (see main manuscript) using the saturation water vapor $e_s$ calculated from the NCEP temperature over the North Atlantic domain ($100^\circ W$-$30^\circ E$/$0^\circ N$-$40^\circ N$; Figure SM2). The scatter in the observed trends is greater than in the predicted ones, which is consistent with the wet-day mean also being affected by factors other than $e_s$.

[Figure]

**Figure SM7.** Map of the historical trends in the wet-day mean $\mu$ in the period 1961-2014. The trend is generally increasing, but there are a few stations showing a decrease. These outliers are probably spurious, as they do not match the bulk of the data.

**Trend in wet−day frequency (1961−2014)**

Figure SM8

**Figure SM8.** Trend estimates of the wet-day frequency $f_w$ for the 1032 locations for the period 1961-2014 suggests values scattered around zero. The cluster of trend values around zero is consistent with the annual wet-day frequency being stationary, but there are regions with significant trends (Figure SM9).

[Figure]

**Figure SM9.** Map of the historical trends in the wet-day frequency $f_w$ for the period 1961-2014. There has been a general increase in the number of wet-days in southern Scandinavia but otherwise no coherent pattern.

[Figure]

**Figure SM10.** Scatter plot showing the correlation between the climatological mean daily maximum temperature (converted to saturation vapour pressure) and the wet-day mean $\mu$. The size of the symbols is proportional to the number of rainy days. The inset shows locations of stations used to compare the climatological mean wet-day mean against the mean surface temperature. The colours of symbols in the scatter plot match those in the map. The data included CLARIS data set from South America, a subset of the ECA&D in Europe used in the COST-VALUE experiment 1, and a subset of station data from GDCN as in Smith et al. (2015) but selecting the stations with the longest records. The selection of location was also limited to sites where both temperature and precipitation had been recorded. Only stations with more than 20000 valid data points were selected, and only the 1945–2015 period was used.

[Figure]

Figure SM11

**Figure SM11.** Comparison between the regression coefficients estimated for each location based on the seasonal cycles in $\mu$ and $e_s$ (blue) and based on the regression analysis of the mean climatology of $\mu$ and $e_s$ at various stations in Europe, South America and North America as in Figure SM10 (grey). Error bars represent two standard errors. The size of the symbols is proportional to the $R^2$ statistics from the regression analysis between the two mean seasonal cycles. The comparison between the results from the two types of analyses suggests a consistency within the margin of error for the locations where the mean seasonal cycle in $\mu$ matched that of the regionally averaged $e_s$ in the predictor domain (Figure SM2).